# Platelet-Derived PDGFB Promotes Recruitment of Cancer-Associated Fibroblasts, Deposition of Extracellular Matrix and Tgfβ Signaling in the Tumor Microenvironment

**DOI:** 10.3390/cancers14081947

**Published:** 2022-04-12

**Authors:** Yanyu Zhang, Ehsan Manouchehri Doulabi, Melanie Herre, Jessica Cedervall, Qi Qiao, Zuoxiu Miao, Anahita Hamidi, Lars Hellman, Masood Kamali-Moghaddam, Anna-Karin Olsson

**Affiliations:** 1Department of Neurosurgery, Xijing Hospital, Fourth Military Medical University, Xi’an 710032, China; yanyu.zhang@fmmu.edu.cn; 2Department of Medical Biochemistry and Microbiology, Science for Life Laboratory, Biomedical Center, Uppsala University, SE-75123 Uppsala, Sweden; melanie.herre@imbim.uu.se (M.H.); jessica.cedervall@imbim.uu.se (J.C.); qiaoqi19950928@outlook.com (Q.Q.); miaozuoxiu@163.com (Z.M.); anahita.hamidi@cnic.es (A.H.); 3Department of Immunology, Genetics and Pathology, Science for Life Laboratory, Uppsala University, SE-75108 Uppsala, Sweden; ehsan.manouchehri@igp.uu.se (E.M.D.); masood.kamali@igp.uu.se (M.K.-M.); 4Department of Cell and Molecular Biology, Biomedical Center, Uppsala University, SE-75124 Uppsala, Sweden; lars.hellman@icm.uu.se

**Keywords:** platelets, PDGFB, extracellular matrix, TGFβ, proximity extension assay (PEA)

## Abstract

**Simple Summary:**

The aim of this study was to investigate the relative contribution of PDGFB derived specifically from platelets to the remodeling of the extracellular matrix (ECM) in tumors. Platelets are a major source of growth factors, which are released from the platelet granules upon activation. Platelets are continuously activated in the tumor microenvironment, due to their similarities to a wound. However, the role of platelet-derived factors in ECM remodeling has not been fully addressed. To this end, we made use of a mouse model with conditional deletion of PDGFB in platelets, which was crossbred to the RIP1-Tag2 model for pancreatic neuroendocrine carcinoma. The amount of tumor-associated PDGFB protein showed a 10-fold reduction in mice lacking PDGFB in platelets. Moreover, ECM deposition, the amount of cancer-associated fibroblasts and TGFβ signaling were all reduced in tumors from mice with platelet-specific deletion of PDGFB, demonstrating the significant contribution of a platelet-derived factor to ECM remodeling in tumors.

**Abstract:**

Platelets constitute a major reservoir of platelet-derived growth factor B (PDGFB) and are continuously activated in the tumor microenvironment, exposing tumors to the plethora of growth factors contained in platelet granules. To address the specific role of platelet-derived PDGFB in the tumor microenvironment, we have created a mouse model with conditional knockout of PDGFB in platelets (pl-PDGFB KO). Lack of PDGFB in platelets resulted in 10-fold lower PDGFB concentration in the tumor microenvironment, fewer cancer-associated fibroblasts and reduced deposition of the extracellular matrix (ECM) molecules fibronectin and collagen I in the orthotopic RIP1-Tag2 model for pancreatic neuroendocrine cancer. Myosin light chain phosphorylation, promoting cell contraction and, consequently, the mechano-induced release of active transforming growth factor (TGF) β from extracellular compartments, was reduced in tumors from pl-PDGFB KO mice. In agreement, TGFβ signaling, measured as phosphorylated Smad2, was significantly hampered in tumors from mice lacking PDGFB in their platelets, providing a plausible explanation for the reduced deposition of extracellular matrix. These findings indicate a major contribution of platelet-derived PDGFB to a malignant transformation of the tumor microenvironment and address for the first time the role of PDGFB released specifically from platelets in the remodeling of the ECM in tumors.

## 1. Introduction

Platelets are essential players in the hemostatic response upon wounding. Exposure of subendothelial molecules, such as von Willebrand factor, collagen and tissue factor (TF), induce platelet adhesion to the vessel wall through specific cell surface-ligand interactions, which results in platelet activation, aggregation through fibrinogen bridging, and the formation of a primary hemostatic plug. Simultaneous activation of the coagulation cascade generates thrombin, which subsequently cleaves fibrinogen to form a stable fibrin clot that prevents bleeding and contributes to the healing processes [1,2]. Aberrant platelet activation has, however, been connected to pathological conditions such as thrombosis and cancer. In fact, 20% of those experiencing unprovoked venous thromboembolism are diagnosed with a malignancy, and cancer-associated thrombosis is a major clinical issue [3,4].

In the tumor microenvironment (TME), platelets are continuously activated due to the presence of similar activation signals in a wound. The discontinuous tumor endothelium exposing subendothelial compartments containing, for example, collagen, is a strong activation signal for platelets [5]. Deregulated expression of TF, both on tumor cells and the tumor endothelium, results in thrombin generation, a potent platelet activator [6,7]. Indeed, tumors have been described as wounds that do not heal [8]. In addition to increased platelet activation, tumors are characterized by constitutive angiogenesis and inflammation. High stroma production and infiltration of fibroblasts are other similarities between tumors and wounds. An important difference between tumors and wounds is, however, that the activation signal is transient in wounds and ceases when the wound is healed. In contrast, the “healing response” is continuous in the TME, commonly characterized by excessive extracellular matrix (ECM) production and ultimately, generation of a fibrotic microenvironment [8,9].

Despite the awareness of tumors as potent sites for platelet activation, and the abundance of cytokines and other bioactive molecules released upon their activation, the contribution of specific platelet-derived factors to the TME composition has not been fully addressed. Commonly, the impact of platelets has been studied through systemic depletion of the entire platelet population in mice, using anti-Gp1b antibodies. A few studies have been performed, where platelet granule proteins have been specifically deleted from megakaryocytes, and thus platelets [10,11,12]. For example, Labelle et al. showed in 2011 that platelet-derived TGFβ is of crucial importance for the epithelial to mesenchymal transition (EMT) of tumor cells and the subsequent formation of metastases [10]. Braun et al. demonstrated that platelet-derived angiopoietin-1, through endothelial Tie-2 receptor signaling, was required to prevent vascular leaks during neutrophil diapedesis [11]. We have recently generated a megakaryocyte/platelet-specific knockout of PDGFB (pl-PDGFB KO) [12]. By crossbreeding the pl-PDGFB KO mice to the spontaneous RIP1-Tag2 (RT2) model for pancreatic neuroendocrine carcinoma, we identified a previously unknown role for platelet-derived PDGFB in the pericyte recruitment to tumor vessels and maintenance of vascular integrity [12]. An observation we made during that study was that RT2 tumors from pl-PDGFB KO mice displayed a softer structure compared to those from WT mice. The molecular cause of this phenotype was, however, not investigated. Considering the well-described correlation between matrix stiffness and ECM deposition, we set out in the current study to investigate ECM and ECM-related parameters in tumors from WT and pl-PDGFB KO mice.

## 2. Materials and Methods

### 2.1. Mice

The animal work was approved by the local ethics committee (5.8.18–09777/2018) and performed according to the United Kingdom Coordinating Committee on Cancer Research (UKCCCR) guidelines for the welfare of animals in experimental neoplasia [13]. Mice of different genotypes were generated and compared to littermates. Sample size (*n*) is given in each figure legend. Analyses were performed in a blinded fashion where relevant, including quantification of immunohistochemical staining. Randomization was not applicable, as no treatments were performed. All mice used in the study were on a C57BL/6 background. To deplete PDGFB specifically in platelets, *Pf4-Cre* mice [14] (C57BL/6-Tg(Pf4-cre)Q3Rsko, The Jackson Laboratory, Bar Harbor, ME, USA), were crossed with *Pdgfb^fl/wt^* mice [15] (B6.129P2-Pdgfb^tm2Cbet^, The Jackson Laboratory, USA) to generate *Pf4-Cre;Pdgfb^fl/wt^* mice, which were then bred with *Pdgfb^fl/wt^* mice again to acquire the *Pf4-Cre;Pdgfb^fl/fl^* mice. To obtain RIP1-Tag2 positive (RT2) (01XD5, NCI-Mouse Repository, Frederick, MD, USA) mice that lack PDGFB in platelets, *Pdgfb^fl/fl^* mice were crossed with RT2 mice to generate *RT2;Pdgfb^fl/wt^* mice, which were bred with *Pf4-Cre;Pdgfb^fl/wt^* mice to acquire *RT2;Pf4-Cre;Pdgfb^fl/fl^* mice. From 10 weeks of age, RT2-positive mice received drinking water supplied with 10% sucrose, to relieve hypoglycemia. DNA extracted from tail biopsies was used for genotyping by PCR. The following primers were used: forward PF4-Cre 5′-CCC ATA CAG CAC ACC TTT TG-3′; reverse PF4-Cre 5′-TGC ACA GTC AGC AGG TT-3′; forward PDGFB fl 5′-GGG TGG GAC TTT GGT GTA GAG AAG-3′; reverse PDGFB fl 5′-GGA ACG GAT TTT GGA GGT AGT GTC-3′; forward Tag2 5′-GGA CAA CCA CAA CTA GAA TGC AG-3′; reverse Tag2 5′-CAG AGC AGA ATT GTG GAG TGG-3′.

### 2.2. Immunostainings

Cryosections (5 μm) fixed in ice-cold methanol were used for all immunostainings except for collagen 1, where ice-cold acetone was used. Blocking with 3% BSA was performed to prevent unspecific binding. Tissues were counterstained with Hoechst (Molecular Probes, Eugene, OR, USA) after incubation with primary antibody and fluorochrome-conjugated secondary antibodies.

### 2.3. Antibodies

The following antibodies and concentrations were used: anti-collagen 1 (2 mg/mL, Abcam 34710), anti-fibronectin (1 μg/mL, Abcam 2413, Cambridge, MA, USA), anti-PDGFRɑ (10 μg/mL, AF1062, R&D, Minneapolis, MN, USA), anti-p-MLC2 (2 μg/mL, Cell signaling 3671, Danvers, MA, USA), anti-p-Smad2 (10 μg/mL, AB3749-I, Millipore, Burlington, MA, USA), anti-cleaved Caspase-3 (1:250, Cell signaling 9661), anti-Phospho-Histone H3 (1:200 μg/mL, Cell Signaling 9701), anti-Ki67 (1 μg/mL, Abcam 15580), anti-rabbit Alexa 555 (2 μg/mL, A31572, Thermo Fisher Scientific, Waltham, MA, USA), anti-rabbit Alexa 488 (2 μg/mL, A21206, Thermo Fisher Scientific), anti-goat Alexa 555 (2 μg/mL, A21432, Thermo Fisher Scientific), anti-rabbit Alexa 647 (2 μg/mL, A21443, Thermo Fisher Scientific).

### 2.4. Masson Trichrome Staining

The collagen deposition was analyzed by Masson trichrome staining. Cryosections (5 μm) were used and staining was performed according to the instruction of the Trichrome stain kit (ab150686, Abcam).

### 2.5. Tumor RNA Extraction and qPCR

Mice were sacrificed by cervical dislocation. Tumors were dissected and snap-frozen in isopentane with dry ice and stored in a −80 °C freezer. RNA was extracted using the RNeasy.

Midi Kit (Qiagen; 75142, Germantown, MD, USA). cDNA was generated using the iScript cDNA synthesis kit (1708891, Bio-Rad, Hercules, CA, USA) and KAPA SYBR FAST qPCR kit (KK4608, KAPA Biosystems, Wilmington, MA, USA) was used for the PCR reaction. The following primers were used: forward collagen 1 5′-TAA GGG TCC CCA ATG GTG AGA-3′; reverse collagen 1 5′-GGG TCC CTC GAC TCC TAC AT-3′; forward fibronectin 5′-GAT GCC GAT CAG AAG TTT GG -3′; reverse fibronectin 5′-GGT TGT GCA GAT CTC CTC GT-3′; forward PDGFB 5′-GGA GTC GGC ATG AAT CGC T-3′; reverse PDGFB 5′-GCC CCA TCT TCA TCT ACG GA-3′; forward PDGFA 5′-GAT GGT ACT GAA TTT CGC CGC-3′; reverse PDGFA 5′-GGG TAT CTC GGC TTC CTC G-3′; forward PDGFC 5′-ACA TTT GAT GAG AGA GAT TTG GGC T-3′; reverse PDGFC 5′-CAG CGT CCT AAA ACA CTT CCA T-3′; forward PDGFD 5′-AAG CAG CCT CAG AGA GAC CAA C-3′; reverse PDGFD 5′-AGT GAG AGT GGG GTC CGT TA-3′; forward TGFβ 5′-CTT CAA TAC GTC AGA CAT TCG GG-3′; reverse TGFβ 5′-GTA ACG CCA GGA ATT GTT GCT A-3′; forward HPRT 5′-CAA ACT TTG CTT TCC CTG GT-3′; reverse HPRT 5′-TTC GAG AGG TCC TTT TCA CC-3′; PDGFRɑ primers were purchased from Qiagen (QuantiTect Primer assays QT0014002).

### 2.6. Isolation and RNA Extraction of Endothelial Cells from RT2 Tumors

Tumors were dissected and digested with 5 mg/mL of Collagenase II and 50 μg/mL of DNase I (both from Sigma-Aldrich, St. Louis, MO, USA) in DMEM for 40 min at 37 °C in a shaking incubator. Tumor lysates were passed through a 70 μm cell strainer and erythrocyte lysis was performed on ice. The samples were blocked with Fc block (101302, Biolegend, San Diego, CA, USA) and stained with anti-CD31-PE (553373, BD Biosciences, Franklin Lakes, NJ, USA) and anti-CD45-APC (103112, Biolegend) for 40 min. After washing with FACS sorting buffer (2% BSA/1 mM EDTA/PBS), samples were stained with DAPI for live/dead discrimination. Endothelial cells (CD45^−^CD31^+^) were sorted using a FACSaria III (BD Biosciences) and collected in PBS, centrifuged, and RNA was extracted using the NucleoSpin RNA Plus kit (740984, Macherey-Nagel, Düren, Germany).

### 2.7. Proximity Extension Assay

The proximity extension assay (PEA) was carried out as a singleplex version of the multiplex PEA previously described by Assarsson et al. [16] and Shen et al. [17], with some modifications. Briefly, the two PEA probes were prepared by diluting 20 µg of rabbit anti-PDGFB antibodies (SAB4502136, Sigma-Aldrich) to 1 µg/µL in 1 × phosphate-buffered saline (PBS). The antibodies were mixed with 33.3-fold molar access of 25 mM of dibenzylcyclooctyne NHS (DBCO-NHS) ester (761524, Sigma-Aldrich) dissolved in 4 mM of DMSO and incubated for 30 min at 25 °C. The activated antibodies were purified using 7K MWCO Zeba Spin columns (10056033, Thermo Fisher Scientific) according to the manufacturer’s instructions. The purified and DBCO-modified antibodies were then divided into two aliquots. Each aliquot was mixed with a 2.5-fold molar excess of 5′-azide-GAG TTT ATA CGG GAA AGT TCA TGG AAT CGA GCC GAC TCG CTT GAA CCT ATG ACT GCA CCT TAT GCT ACC GTG ACC TGC GAA TCC AGT CT-3′ and 5′-Azide-CCA CTG GGT CTG GTC AAT CAC GCG GCG GCA TGT GAA TAG TAG ATC ACG ATG AGA CTG GAT GAA-3′ (Integrated DNA Technologies, Coralville, IA, USA), respectively, and incubated overnight at 4 °C. The DNA oligonucleotide conjugated antibodies were validated on a 10% TBE urea denaturing gel, and stored in a storage buffer (1 × PBS, 0.1% BSA, 0.05% NaN_3_) at 4 °C until use.

A two-fold dilution series of pure PDGFB antigen (from Mouse/Rat PDGF-BB DuoSet ELISA kit; DY8464–05, R&D systems) from 2 ng/mL to 62.5 pg/mL was prepared in sample diluent (84032, Olink Proteomics, Uppsala, Sweden). The pair of PEA probes were mixed in the antibody diluent buffer (1 mM of D-Biotin, 0.05% Tween 20, 1% BSA in 1 × PBS, 0.1 mg/mL of salmon sperm DNA (15632, Invitrogen, Waltham, MA, USA), 0.1 µM of goat IgG (I5256, Sigma-Aldrich), 1 × EDTA in ultra-pure H_2_O) at a final concentration of 100 pM each. One μL of serial dilution samples or tissue lysates was mixed with 3 μL of probe mixture and incubated for 1 h at 37 °C. Thereafter, 96 μL of PEA extension mix (10 mM of Tris-HCl (pH 8.5, 25 °C), 50 mM of KCl, 0.075% TritonX-100, 1.5 mM of MgCl_2_, 1.25% DMSO, 1.25% Glycerol, 0.5 mM of Tris-HCl (pH 8.5), 0.02 mM of dNTP mix (R0182, Thermo Fisher Scientific), 1 μM each of PCR forward primer: 5′-CAC GAC TCT AGC ATG TCT ACG-3′ and PCR reverse primer: 5′-CGC AGT TAA TGT GAT ATG GCC-3′ (Integrated DNA Technologies) in ultra-pure H_2_O, 1 U OneTaq Hot Start (M0481L, NEB) was added. The extension and first amplification steps were performed with the following cycles: 50 °C for 20 min and 95 °C for 5 min, followed by 17 cycles of 95 °C for 30 s, 54 °C for 1 min and 60 °C for 1 min. Thereafter 2.5 μL of the PCR product was mixed with 7.5 μL of the PCR master mix (10 × PCR Buffer (10966034, Invitrogen), 2.5 mM of MgCl_2_, 0.5 × sybr green I, 0.25 mM of dNTPs, 0.03 U/µL of Platinum Taq Polymerase (10966034, Invitrogen), 0.9 μM of each forward and reverse PCR primers, ROX (12223–012, ThermoFisher), PCR-graded H_2_O). The qPCR was performed on a QuantStudio 6 instrument using the following cycles: 95 °C for 2 min followed by 45 cycles of 95 °C for 15 s and 60 °C for 1 min.

### 2.8. Image Analysis

Imaging of tissue sections stained by immunofluorescence or Masson trichrome was done using a Nikon Eclipse 90i microscope and the NIS Elements 3.2 software. Images were analyzed using the Image J 1.45s software (National Institute of Health, Bethesda, MD, USA).

### 2.9. Statistics

Statistical analyses were performed using the non-parametric two-tailed Mann–Whitney test when normality could not be assumed. When normality could be assumed, the parametric two-tailed Student’s *t*-test was used. In this study, *n* always represents individual samples. * is defined as *p* ≤ 0.05, ** as *p* ≤ 0.01, *** as *p* ≤ 0.001 and **** as *p* < 0.0001. Exact *p* values are given in the respective figure legend.

## 3. Results

### 3.1. Platelet-Specific Ablation of PDGFB Reduced Extracellular Matrix Formation in the Tumor Microenvironment

Mice lacking PDGFB in the megakaryocyte/platelet lineage (pl-PDGFB KO) were generated by crossbreeding *Pf4-Cre* mice with *Pdgfb^fl/fl^* mice (Figure 1A and [12]). Pl-PDGFB KO mice are viable and fertile, with no phenotypical or functional defects in the megakaryocyte or platelet population [12]. To study the specific contribution of platelet-derived PDGFB to the tumor microenvironment (TME), pl-PDGFB KO mice were crossbred to the RIP1-Tag2 (RT2) model for pancreatic neuroendocrine carcinoma [12]. The RT2 model develops in a synchronous and stepwise fashion through hyperplasia and dysplasia, the angiogenic switch, local cancer, and finally, invasive carcinoma with spontaneous metastasis, primarily to the liver [18]. Primary tumor growth was not affected by the lack of platelet-derived PDGFB. In contrast, the amount of circulating tumor cells, as well as the number of liver metastases, were significantly enhanced as a result of a compromised vascular barrier in pl-PDGFB KO mice [12]. To address if the composition of the extracellular matrix (ECM) in tumors from pl-PDGFB KO mice differs from WT mice, we performed immunostainings for collagen 1 (Col1), the main ECM component in connective tissue [19]. We detected a reduced amount of Col1 in tumors from pl-PDGFB KO mice compared to WT (Figure 1B,C). This finding was confirmed using Masson-trichrome staining detecting collagen fibers. (Appendix A). Immunostaining for fibronectin (FN), another major component of the ECM, also showed a reduced signal and a somewhat distorted and less elongated pattern in tumors from mice lacking PDGFB in their platelets (Figure 1D,E). To address if the reduced immunoreactivity of Col1 and FN in tumors from pl-PDGFB KO mice was due to decreased gene expression, qPCR analysis was performed using RNA isolated from the respective tumors. There was a significant reduction in Col1 transcription in tumors from pl-PDGFB KO mice compared to WT (Figure 1F), while no difference was detected with respect to FN (Figure 1G).

### 3.2. The Amount of PDGFB Is Reduced in Tumors from pl-PDGFB KO Mice

When platelets are activated in response to an injury or in the TME, they leave the circulation, adhere and release their granule content, including PDGFB (Figure 2A). To compare the amount of PDGFB in tumors from WT and pl-PDGFB KO mice, we established a sensitive PEA, allowing detection of low abundant PDGFB in tumor tissue lysates. As illustrated in Figure 2B, there was an approximately 10-fold reduction in the PDGFB concentration detected by PEA in tumor lysates from pl-PDGFB KO mice compared to that from WT mice (Figure 2B). This experiment shows, for the first time, the relative contribution of PDGFB derived specifically from platelets in the TME, compared to other cellular sources. At the RNA level, there was no difference between tumors from WT and pl-PDGFB KO mice with respect to PDGFB expression (Figure 2C), demonstrating that the reduced level of tumor-associated PDGFB in pl-PDGFB KO mice was not due to changes in the transcriptional level in cells residing in the TME. Overall, PDGFB expression detected using RNA extracted from whole RT2 tumors was low compared to the expression in endothelial cells (ECs) isolated from RT2 tumors (Appendix A). This indicates that PDGFB is not expressed by the actual tumor cells in this model and that ECs are the major source of PDGFB transcription in the TME.

### 3.3. Tumors in pl-PDGFB KO Mice Have Fewer Cancer-Associated Fibroblasts

Fibroblasts respond to PDGFB through their expression of PDGFRα and/or β. PDGF-dependent signaling promotes both the proliferation and differentiation of fibroblasts into cancer-associated fibroblasts (CAFs) [20,21]. In response to TGFβ, commonly present in large amounts in the TME, CAFs can express high levels of ECM proteins such as collagens and fibronectin [22]. To investigate if the amount of CAFs was altered in pl-PDGFB KO mice compared to WT, we performed immunostainings for PDGFRα in RT2 tumors (Figure 3A). Quantification revealed a significant reduction in the PDGFRα^+^ area by more than 50% (Figure 3B). This finding was supported by qPCR on tumor RNA showing a similar reduction in PDGFRα transcripts in pl-PDGFB KO mice compared to WT (Figure 3C). These data indicate a reduced amount of CAFs in tumors from mice lacking PDGFB in their platelets. To address if the reduced amount of CAFs in tumors from pl-PDGFB KO mice could be attributed to changes in proliferation or apoptosis, we performed co-immunostaining for PDGFRα in combination with either phospho-histone 3 (pH3), to assess proliferation, or cleaved Caspase-3, to assess apoptosis. We detected a strong trend for the reduced proliferation of the PDGFRɑ+ cells in tumors from pl-PDGFB KO mice, although not statistically significant at this time point (Appendix A). There was also no significant difference in the number of apoptotic cells in tumors of the two genotypes (Appendix A).

A feature of CAFs is their ability to take on a contractile phenotype [23]. Cell contraction is dependent on the action of myosin, which in an ATP-dependent process can drive actin filament sliding, and thus exert a pulling force on the ECM where the cell is anchored through integrins [24]. In non-muscle cells, contraction is primarily regulated by phosphorylation of one of the myosin light chains (MLC) by myosin light chain kinase (MLCK) [25]. Immunostaining for phosphorylated MLC (p-MLC) revealed a significant reduction in tumors from pl-PDGFB KO mice compared to WT (Figure 3D,E), indicating a reduced contractile activity in the TME of mice lacking platelet PDGFB. Moreover, the PDGFRα and p-MLC staining largely overlapped (Figure 3F), demonstrating that the contractile activity in the RT2 tumor microenvironment is mainly exerted by CAFs.

### 3.4. TGFβ-Dependent Signaling Is Reduced in Tumors from pl-PDGFB KO Mice

Signaling through the TGFβ pathway is a major inducer of ECM production. TGFβ is secreted in an inactive form bound to the latency-associated peptide (LAP), which mediates the deposition of TGFβ in the ECM through its interaction with microfibrils [22,26,27]. Release of active TGFβ from this complex requires either proteolytic processing or the mechano-induced stretching of LAP through integrin binding to the RGD-domain in LAP [27,28]. To address if TGFβ signaling was affected in tumors from pl-PDGFB KO mice, we performed immunostaining for phosphorylated Smad2 (p-Smad2), a downstream target of TGFβ receptor signaling. Abundant p-Smad2 positive cells were detected in tumors from WT mice (Figure 4A). In contrast, p-Smad2 immunoreactivity was not as frequent in tumors from pl-PDGFB KO mice and was significantly reduced compared to WT (Figure 4A,B), demonstrating reduced TGFβ-mediated signaling in the TME in mice lacking PDGFB in their platelets. No change in TGFβ expression was detected in tumors from pl-PDGFB KO mice by qPCR (Appendix A). In addition, no changes in expression of either PDGFA, C or D were detected (Appendix A).

A schematic model based on the findings in this study is provided in Figure 4C. PDGFB derived from platelets activated in the TME, for example, due to the leaky vasculature, contributes to the recruitment of CAFs and thereby increased the expression of Col1. The more Col1 that is deposited, the stiffer the ECM becomes, and consequently more active TGFβ will be released from its extracellular stores due to increased integrin-dependent tension exerted on the ECM.

## 4. Discussion

In the current study, we present for the first time, in vivo evidence for a connection between platelet-derived PDGFB and ECM deposition in the tumor microenvironment. Tumors from mice lacking PDGFB in their platelets display a 10-fold reduction in the amount of PDGFB protein present in the TME, fewer PDGFRα^+^ CAFs, reduced contractile activity (p-MLC) and attenuated TGFβ-dependent signaling (p-Smad2), ultimately resulting in less Col1 expression and FN polymerization. These findings highlight that factors derived from activated platelets can make a significant contribution to the continuous remodeling of the TME.

In this context, it is important to stress that the platelet-specific KO of PDGFB does not affect the megakaryocyte phenotype or numbers, platelet numbers, the ability of platelets to become activated following external activation by ADP or thrombin or the accumulation of platelets in the tumor microenvironment, compared to WT mice [12].

The TME is commonly characterized by excessive ECM production, ultimately generating a fibrotic microenvironment [9]. TGFβ, often expressed at high levels by tumor cells, is a major inducer of ECM production, especially Col1 [22]. This process is creating a vicious circle (Figure 4C); the stiffer the ECM gets, the stronger contractile force the cells in the tumor will exert on their surroundings. This contraction will in turn release an active transforming growth factor β (TGFβ), which is stored in an inactive conformation in the ECM, bound to the inhibitory peptide LAP. Since TGFβ stimulation is a potent inducer of ECM production, this further adds to the stiff microenvironment. Increased ECM stiffness of solid tumors is often associated with malignant progression and poor patient outcomes [29]. Our findings show that platelet-derived PDGFB can modulate the level of TGFβ-dependent signaling in vivo in the TME, likely through supporting recruitment and/or differentiation of fibroblasts to ECM-producing CAFs. It is also possible that PDGFB and TGFβ act synergistically at the transcriptional level to promote Col1 expression. In this context, it is important to note that the amount of TGFβ released from activated platelets is not disturbed in pl-PDGFB KO mice [12]. Neither was there any change in TGFβ expression detected in tumors from pl-PDGFB KO mice. We, therefore, conclude that the reduced p-Smad2 level in tumors from mice with PDGFB-deficient platelets is most likely due to the lower release of TGFβ from extracellular stores.

Cancer-associated fibroblasts (CAFs) are major producers of ECM proteins, such as Col1 and fibronectin, and constitute a heterogeneous cell population in the TME [23]. There is no pan-CAF-marker that can be used to identify all CAFs. Instead, a number of CAF subtypes have recently been identified using single-cell RNA sequencing, for example, myofibroblast-like CAFs, expressing large amounts of ECM proteins, and inflammatory CAFs, with an inflammatory cytokine profile [30]. PDGFRα expression, which we find downregulated in the TME of pl-PDGFB KO mice, has been reported in both inflammatory CAFs and ECM-producing CAFs [23].

The mechanism by which the reduced PDGFB level in the TME of pl-PDGFB RT2 tumors resulted in fewer CAFs, could potentially involve effects on proliferation, migration or survival, or all of these processes. We did not detect a significant difference in either the proliferation or apoptosis of CAFs in tumors derived from pl-PDGFB KO or WT mice. However, there was a strong trend for the reduced proliferation of CAFs in the pl-PDGFB KO mice and it is possible that such a difference could lead to a reduced number of CAFs over time. Considering the potent chemotactic effect PDGFB exerts on fibroblasts in vitro [31,32,33], it is also fully possible that the reduced amount of CAFs is caused by impaired recruitment. Moreover, we could not detect any compensatory upregulation of other PDGF isoforms in the TME of pl-PDGFB KO mice.

There are previous publications describing the role of PDGFB/PDGFRs in tumor ECM remodeling. Transgenic mouse strains overexpressing PDGFA, -B, -C or -D all develop liver fibrosis [34]. In agreement, Hammer et al. [35] demonstrated that hyperactivation of the PDGFRɑ in mammary fibroblasts resulted in fibrosis and increased stiffness of the mammary tissue. Moreover, tumor cells injected orthotopically in the mammary fat pad of these mice grew larger tumors compared to controls [35]. The PDGFRβ has also been shown to regulate interstitial fluid pressure in tumors through ECM remodeling. In preclinical colon carcinomas, treatment with a PDGFRβ kinase inhibitor decreased interstitial hypertension and improved capillary-to-interstitium transport [36].

While the reduced Col1 fibrillar network in tumors from pl-PDGFB KO mice was paralleled by the downregulation of Col1 also on the transcriptional level, FN gene expression was not altered despite the reduced FN network formation in tumors from mice lacking PDGFB in the platelets. A possible explanation lies in the mechanism by which FN is polymerized to form a network. FN polymerization is dependent on binding to integrins on adherent cells, which in turn exerts a pulling force on the FN molecule. This unfolds the FN molecule and exposes binding sites for additional FN molecules, which is a prerequisite for FN network formation [37]. The more collagen in the ECM, the stiffer matrix and the stronger pulling force can be exerted by the cells on the FN molecule, resulting in more FN fibrillogenesis. A reduction in the FN network in the pl-PDGFB KO tumors can thus be a consequence of the reduced matrix stiffness due to lower Col1 expression.

The PDGF/PDGFR signaling axis has a prominent and well-established role in tumor progression [38,39]. Traditionally, the focus has been on PDGF expressed by the tumor cells in this context. Our data show that PDGFB derived from platelets also has a significant impact on the remodeling of the TME. In addition to PDGFB, platelets constitute a major reservoir for a number of growth factors and cytokines with a strong impact on the TME, such as vascular endothelial growth factor (VEGF) and TGFβ [40,41]. This fact is important to take into consideration when analyzing cytokine expression in different cancers at the transcriptional level. It may be of equal importance to address the platelet-activating capacity of the specific tumor type, and consequently, the cytokine profile at the protein level, to get a comprehensive picture of the signaling networks and heterotypic interactions that are active in the tumor microenvironment. The data presented in this study shows that the PDGFB concentration in the TME is reduced 10-fold when genetically deleted from the platelet population, demonstrating the large and significant contribution from this cell type to the total pool of PDGFB, and most likely, many other cytokines, in tumor tissue. Proteomic analyses will therefore be an important complement to genomics and transcriptomics, especially in tumors where platelets are continuously activated in the microenvironment.

## 5. Conclusions

In conclusion, our data show that PDGFB derived specifically from platelets plays an important role in ECM deposition and TGFβ-dependent signaling in the TME. The impact of PDGFB and other platelet-derived cytokines in different pathologies warrants further investigation and platelet-targeting therapies may represent an interesting therapeutic strategy, for example, in fibrosis.

## Figures and Tables

**Figure 1 cancers-14-01947-f001:**
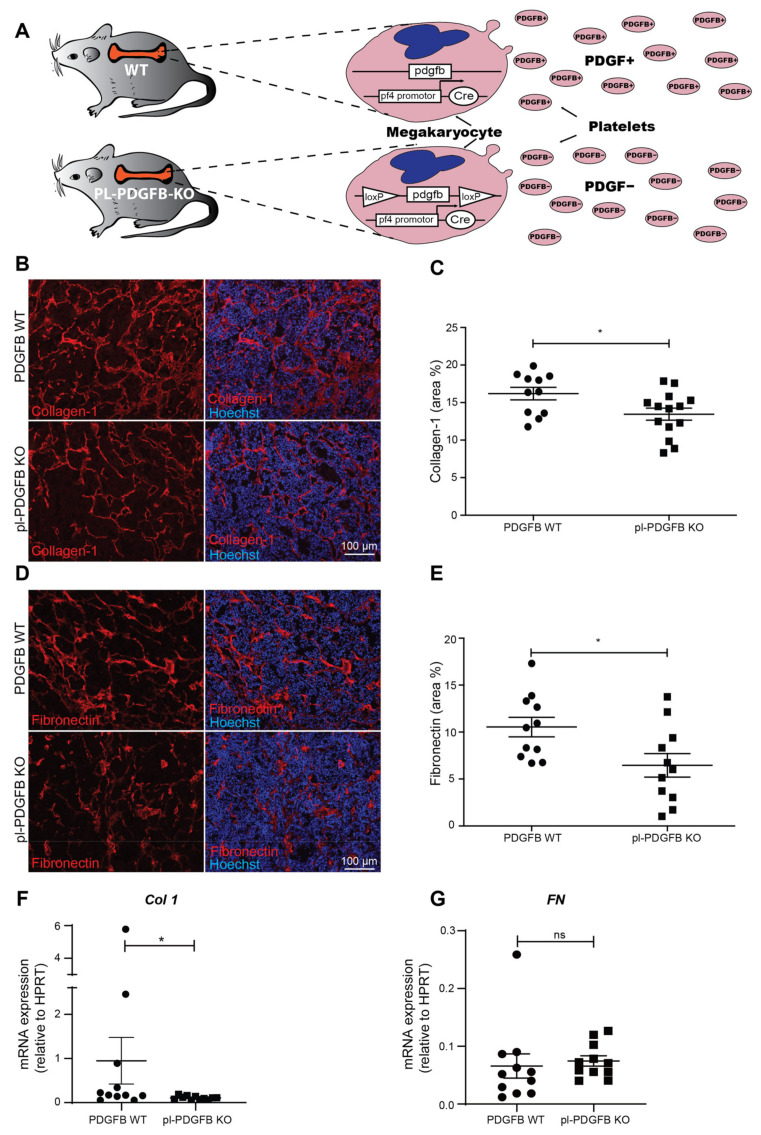
Reduced ECM deposition in tumors from mice lacking PDGFB in platelets. (**A**) Mice lacking PDGFB in the platelet/megakaryocyte lineage were generated using the Cre-loxP recombination system, with Cre expressed under the PF4 promoter. (**B**–**E**) Sections of tumor tissue from 14-week old WT and pl-PDGFB KO RT2-positive mice were immunostained for (**B**,**C**) collagen 1 (WT *n* = 11; KO *n* = 14, * *p* = 0.0280) and (**D**,**E**) fibronectin (WT *n* = 11; KO *n* = 11, * *p* = 0.0207) and the positive areas quantified using Image J. Expression of collagen 1 (Col1;WT *n* = 11; KO *n* = 11, * *p* = 0.0336) (**F**) and fibronectin (FN; WT *n* = 11; KO *n* = 11, * *p* = 0.1464) (**G**) were analyzed by qPCR in tumors from 14-week old WT and pl-PDGFB KO RT2-positive mice. Error bars in the graphs represent the standard error of mean (SEM).

**Figure 2 cancers-14-01947-f002:**
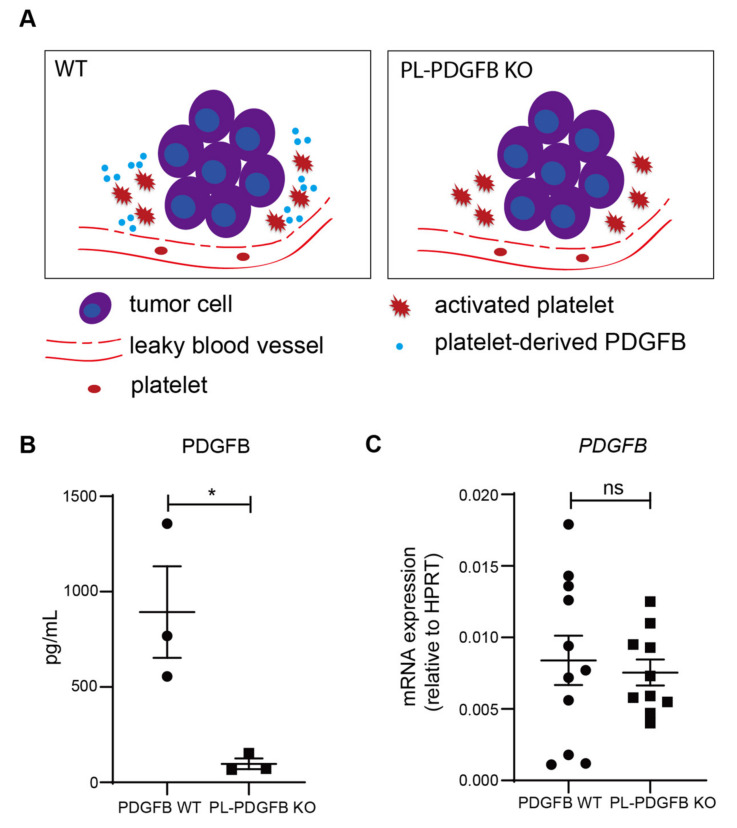
Reduced PDGFB levels in tumors from mice lacking PDGFB in platelets. (**A**) PDGFB is released from activated platelets in the TME in WT mice, but not in pl-PDGFB KO mice. (**B**) The amount of PDGFB protein in tumors derived from 14-week old WT and pl-PDGFB KO RT2-positive mice was analyzed using PEA in tissue lysates from whole tumors (WT *n* = 3; KO *n* = 3, * *p* = 0.0300). (**C**) Expression of PDGFB was analyzed by qPCR using RNA extracted from whole tumors from 14-week old WT and pl-PDGFB KO RT2-positive mice (WT *n* = 11; KO *n* = 10, *p* = 0.6776 (ns)).

**Figure 3 cancers-14-01947-f003:**
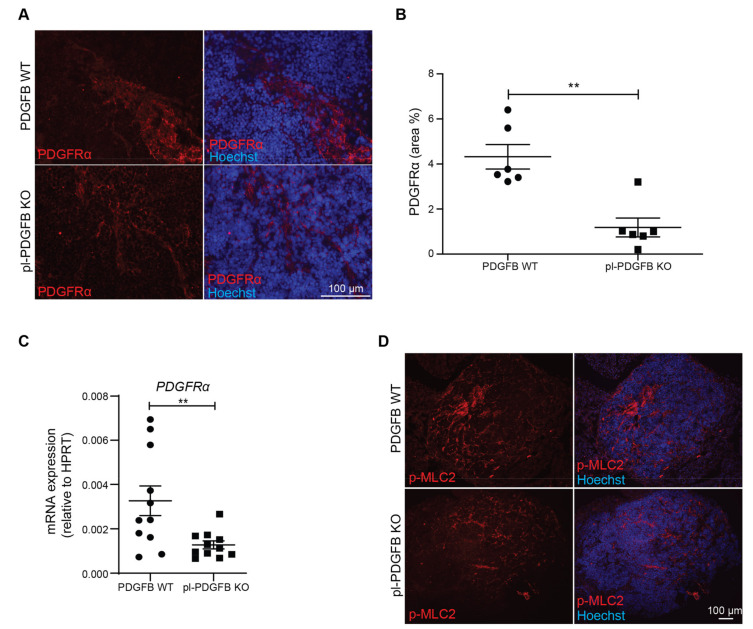
Fewer cancer-associated fibroblasts and reduced myosin light chain (MLC) phosphorylation in tumors from pl-PDGFB KO mice. (**A**,**B**) Tumor sections from 14-week old WT and pl-PDGFB KO RT2-positive mice were immunostained for PDGFRα (*n* = 6/group, ** *p* = 0.0022) and the positive area quantified using Image J. (**C**) Expression of PDGFRα at the transcriptional level was analyzed by qPCR using RNA extracted from the same type of tumors as in panel A and B (*n* = 11/group, ** *p* = 0.0095). (**D**,**E**) Tumor sections from 14-week old WT and pl-PDGFB KO RT2-positive mice were immunostained for phosphorylated MLC2 (*p*-MLC2) (WT *n* = 7; KO *n* = 8, * *p* = 0.0184), and the positive areas quantified using Image J. (**F**) Co-staining of PDGFRα and p-MLC was performed on tumor tissue from 14-week old RT2-positive mice WT and pl-PDGFB KO mice. Error bars in the graphs represent the standard error of mean (SEM).

**Figure 4 cancers-14-01947-f004:**
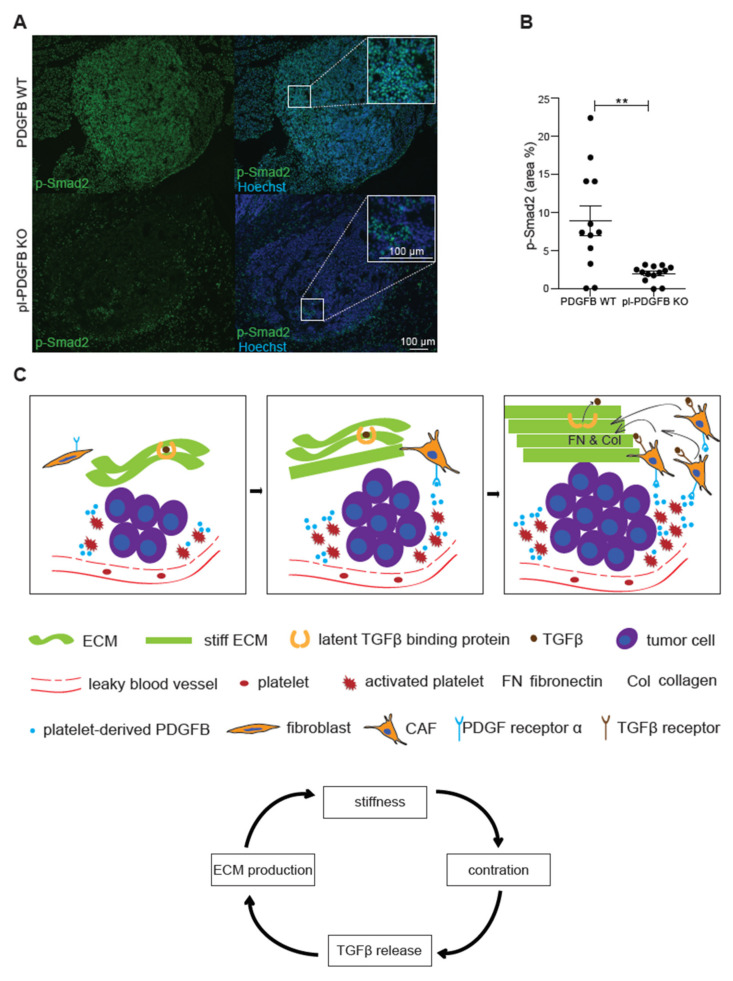
Reduced TGFβ signaling in tumors from mice lacking PDGFB in platelets. (**A**,**B**) Tumors from 14-week old WT and pl-PDGFB KO RT2-positive mice were immunostained for p-Smad2 and the positive areas were quantified using Image J (WT *n* = 12; KO *n* = 13, ** *p* = 0.0015). Error bars in the graphs represent the standard error of mean (SEM). (**C**) Schematic model: the discontinuous endothelium in a tumor exposes platelets to subendothelial spaces, leading to activation, degranulation and release of PDGFB and other platelet-derived molecules in the tumor microenvironment (left panel). The PDGFRα+ CAFs are recruited by PDGFB derived from platelets and other sources in the tumor microenvironment. CAFs express large amounts of ECM proteins such as Col1 and FN, which leads to increased stiffness of the ECM (middle panel). Increased cell contraction due to the stiffer ECM leads to more release of active TGFβ from its extracellular stores, which in turn leads to more ECM expression and subsequently more FN polymerization (right panel). This process is creating a vicious circle (the arrow flow chart at the bottom): the more ECM production, the stiffer matrix and more release of active TGFβ, a potent inducer of ECM expression.

## Data Availability

The data presented in this study are available in the article and Appendix A.

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
