# Peer review of "Platelet-Derived PDGFB Promotes Recruitment of Cancer-Associated Fibroblasts, Deposition of Extracellular Matrix and Tgfβ Signaling in the Tumor Microenvironment"

_cancers, 2022, doi:10.3390/cancers14081947_

Round 1
Reviewer 1 Report
The authors addressed most of the concerns. Some minor comments to improve the manuscript:
Figure 2 legend title should exclude "signaling" as the figure addresses the presence or absence of PDGFB in WT vs. KO mice.
Supplementary Figures 3 and 4 legends are missing.
Figure 4 C schematic should include Collagen in the middle panel as described in the legend.
In method: Statistics section: The sentence mentioning the tests used can be broken into two sentences for more clarity.
Author Response
Response to reviewer’s comments
Reviewer 1
The authors addressed most of the concerns. Some minor comments to improve the manuscript:
Figure 2 legend title should exclude "signaling" as the figure addresses the presence or absence of PDGFB in WT vs. KO mice.
REPLY: We thank the reviewer for noticing this. The word “signaling” has now been replaced by “levels”.
Supplementary Figures 3 and 4 legends are missing.
REPLY: We apologize for this mistake, which has now been corrected.
Figure 4 C schematic should include Collagen in the middle panel as described in the legend.
REPLY: To harmonize Fig 4C with the legend, the figure legend has been modified and now reads “CAFs express large amounts of ECM proteins such as Col1 and FN,…“. We believe this will clarify that ECM molecules (indicated in green) includes both FN and Col1.
In method: Statistics section: The sentence mentioning the tests used can be broken into two sentences for more clarity.
REPLY: We have followed the reviewer’s advice and divided the sentence in two.
Reviewer 2 Report
My remarks have been taken into account, therefore I recommend it for publication
Author Response
Response to reviewer’s comments
Reviewer 2
My remarks have been taken into account, therefore I recommend it for publication
Reviewer 3 Report
Figure S1 and S2 are missing. The legend of figure S3 and S4 need be added into the part of supplementary materials (page 14).
Author Response
Response to reviewer’s comments
Reviewer 3
Figure S1 and S2 are missing. The legend of figure S3 and S4 need be added into the part of supplementary materials (page 14).
REPLY: We apologize for these mistakes, which have now been corrected.
This manuscript is a resubmission of an earlier submission. The following is a list of the peer review reports and author responses from that submission.
Round 1
Reviewer 1 Report
In this manuscript, the authors aimed to identify the mechanisms underlying the development of softer tumors in a spontaneous RIP1-Tag2 (RT2) model for pancreatic neuroendocrine carcinoma that was made platelet-derived growth factor B (PDGFB) deficient specifically in platelets (pl-PDGFB KO RT2). The authors detected reduced presence of extracellular matrix components including collagen 1 and fibronectin in the pl-PDGFB KO RT2 tumors compared to pl-PDGFB WT RT2 tumors. Using proximity extension assay (PEA), the authors found remarkable decrease in PDGFB concentration in the tumor lysates from KO model compared to WT model. The tumor bearing KO model also had a significantly decreased presence of PDGFRα expressing cancer associated fibroblasts (CAFs). Further, the tumors in the KO model had decreased contractile activity, indicated by reduced presence of contractility regulator phosphorylated-myosin light chain (p-MLC). In addition, the authors found less frequent presence of phospho-Smad2 in the tumors from the KO model suggesting a defect in TGF- β signaling pathway activation in the absence of platelet specific PDGFB. Finally, the authors proposed a model, where platelet specific PDGFB contributes to CAF recruitment, which potentially leads to collagen deposition and stiffer ECM formation. Stiffer ECM potentially increases the release of TGF-β from the matrix due to increased contractile force. Released active form of TGF-β feeds into the cycle of ECM component production and increase in matrix stiffness.
The rationale for the present study as stated in the last section of the introduction is to pursue the mechanisms underlying changes in tumor stiffness due to the deficiency of PDGFB in platelets. However, a quantitative comparison of tumor stiffness was not made in the manuscript nor it could be found in the published study (Cancer Research) from the same group. Even though the current study presents an interesting novel role of platelet specific PDGFB in modulating ECM in vivo, to what extent the changes in ECM correlate with changes in stiffness needs to be addressed. Additional studies must be conducted to conclude the current title and propose the model. Following are my major concerns:
- As mentioned above, ECM stiffness needs to be compared between pl-PDGFB KO RT2 and pl-PDGFB WT RT2 tumors. Following review with technical information for quantification might be useful.
10.1021/acsbiomaterials.0c01530
- One important aspect that has not been thoroughly addressed is how PDGFB impacts CAFs presence. The authors detected decreased presence of CAFs in the pl-PDGFB KO RT2 Additional functional assays need to be conducted to conclude whether the reduction is due to direct effect of PDGFB on CAF migration or via other effects such as CAF proliferation, survival, or all of the above processes. The authors in earlier publication (Cancer Research) identified a role of platelet specific PDGFB in pericyte recruitment to tumor vessels. Is it possible for those recruited perciytes to transition into CAFs in their model? (10.1073/pnas.1608384113)
- Does PDGFB impact ECM release from cancer cells or CAFs or both?
- Decrease in TGF- β signaling as indicated by reduced Smad-2 immunofluorescence staining in the pl-PDGFB KO RT2 tumors may also result from an overall decrease in the secretion of TGF- β from cells. Both cancer cells and CAFs, which are known producers of TGF- β need to be tested for their ability to express/secrete TGF- β when isolated from pl-PDGFB KO RT2 vs. WT tumors.
Reviewer 2 Report
The manuscript is interesting and contains new findings. The article is well written but needs some minor corrections:
Figures 1 B and D; 3 A, D and F, and 4 A do not include the information that MERGE stands for Merge with Hoechst
Reviewer 3 Report
PDGF isoforms with their receptors play vital roles in tumorigenesis and cancer development, especially targeting on pericytes, fibroblasts and myofibroblasts of the stroma of solid tumors. Using platelet derived-PDGFB KO animal model, authors illustrates the novel role of pl-PDGFB in pancreatic cancer ECM remodeling with reduced CAFs number and ECM molecules expression. The manuscript is well-written and structured.
Specific comments:
1. Authors should add the essential information about the effect of pl-PDGFB deletion on cancer growth and metastasis in their animal model.
2.The references about tumor ECM and tissue remolding regulated by PDGFB/PDGFRs should be cited and discussed.
3. Is there any effect on platelet accumulation by KO pl-PDGFB in RT2 tumor? The expression of other PDGF isoforms, such as PDGFA, should be tested in pl-PDGF null tumor. It will provide the useful information about whether compensated PDGF isoforms would relief the stress of PDGFB depletion on the ECM remodeling.
